# Neighbourhood Socioeconomic Processes and Dynamics and Healthy Ageing: A Scoping Review

**DOI:** 10.3390/ijerph19116745

**Published:** 2022-05-31

**Authors:** Cláudia Jardim Santos, Inês Paciência, Ana Isabel Ribeiro

**Affiliations:** 1EPIUnit-Instituto de Saúde Pública, Universidade do Porto, 4050-600 Porto, Portugal; ana.isabel.ribeiro@ispup.up.pt; 2Laboratório para a Investigação Integrativa e Translacional em Saúde Populacional (ITR), 4050-600 Porto, Portugal; 3Center for Environmental and Respiratory Health Research (CERH), University of Oulu, P.O. Box 5000, 90570 Oulu, Finland; inespaciencia@gmail.com; 4Biocenter Oulu, University of Oulu, P.O. Box 5000, 90570 Oulu, Finland; 5Departamento de Ciências da Saúde Pública e Forenses e Educação Médica, Faculdade de Medicina, Universidade do Porto, 4200-319 Porto, Portugal

**Keywords:** older adults, neighbourhood socioeconomic status, neighbourhood socioeconomic deprivation, neighbourhood segregation, gentrification, urban renewal, age-friendly communities

## Abstract

Elderly citizens are concentrated in urban areas and are particularly affected by the immediate residential environment. Cities are unequal and segregated places, where there is an intensification of urban change processes such as gentrification and displacement. We aimed to understand how neighbourhood socioeconomic processes and dynamics influence older people’s health. Three bibliographic databases—PubMed, Web of Science, and Scopus—were used to identify evidence of the influence of neighbourhood socioeconomic deprivation, socio-spatial segregation, urban renewal, and gentrification on healthy ageing. We followed the method of Arksey and O’Malley, Levac and colleagues, the Joanna Briggs Institute, and the PRISMA-ScR. The included studies (*n* = 122) were published between 2001 and 2021. Most evaluated neighbourhood deprivation (*n* = 114), followed by gentrification (*n* = 5), segregation (*n* = 2), and urban renewal (*n* = 1). Overall, older people living in deprived neighbourhoods had worse healthy ageing outcomes than their counterparts living in more advantaged neighbourhoods. Older adults pointed out more negative comments than positive ones for gentrification and urban renewal. As to segregation, the direction of the association was not entirely clear. In conclusion, the literature has not extensively analysed the effects of segregation, gentrification, and urban renewal on healthy ageing, and more quantitative and longitudinal studies should be conducted to draw better inferences.

## 1. Introduction

The global population is ageing rapidly. By 2050, the number of people aged 65 years or over is estimated to be 1.5 billion, which is more than double the number observed in 2020 [1] (p. 3). On average, at the age of 65, a person can expect to live another 17 years, and an additional two years is projected for 2045–2050 [2] (p. 2).

Although the population lives longer, people are not necessarily “living” the added years with good health [3] (p. 116) [4]. Worldwide trends indicate that from 2000 to 2019, the healthy life expectancy (HALE) at birth has increased from 58.3 years to 63.7 years. Nevertheless, considerable improvements are needed as the global life expectancy (LE), which was around 66.8 years in 2000, increased to 73.3 years in 2019, indicating that the gap between LE and HALE is growing.

Yet, there are geographical disparities regarding LE and HALE. The African and European regions score the lowest and the highest LE and HALE, respectively, among the World Health Organization (WHO) regions [5] (p. 15). For example, in 2019, the European Union (EU) life expectancy at birth was stated as 84 years for women and 78.5 years for men [6]. However, the HALE at birth was only 65.1 years for women and 64.2 for men [7]. These numbers indicate that even top-performing countries (e.g., Ireland, Malta, Spain, and Sweden) need adjustment strategies to promote healthier ageing overall and guarantee that the last part of a person’s life is lived without the limitation of illness or disability for as long as possible. Thus, the ongoing demographic changes will bring challenges that need to be urgently addressed to optimise opportunities for healthy ageing.

According to the WHO, healthy ageing is “the process of developing and maintaining the functional ability that enables well-being in older age” [8]. Ageing is an inevitable process that everybody experiences; however, not everyone ages equally. It is possible to observe people of the same chronological age who are either completely “healthy” or face many health problems. The differences observed result from genetic inheritance, environmental exposures, and health behaviours. Although genetic factors influence the complex ageing process, the cumulative impact of individual behaviours and environmental exposures throughout life may be more influential on one’s functional ability [8].

Despite all that is known about the environmental exposures that contribute to diverse health outcomes as we age, there is still room to understand the causal mechanisms that connect one’s environmental neighbourhood with their health. Since older adults spend most of their time in their nearby neighbourhood environment due to physical, emotional, and financial reasons, it is essential to better understand how the neighbourhood environment affects this population’s health [9,10,11].

### 1.1. Theoretical Background

Neighbourhoods are unique geographical areas that possess a set of characteristics that can shape the health of its inhabitants. Therefore, the health of individuals cannot be assessed without considering the inequitable distribution of social opportunities and constraints that people are subjected to that have the power to influence individual health behaviour [12] (p. 97) [13]. As social theorist Max Weber mentioned in his book, “Economy and Society: An Outline of Interpretive Sociology”, life choices and life chances are intrinsically related to one’s social situation; in other words, people’s actions are socially determined. Thus, the role of the neighbourhood socioeconomic context in people’s lives must be explored when discussing health outcomes.

Several mechanisms connect neighbourhoods and health. In addition, there are different types of neighbourhood exposures that interact with each other. These exposures include the environment’s shared social and physical features; the availability of a (un)favourable home, work, and leisure environments; the services to support daily living; sociocultural features; and neighbourhood reputation [14]. For this purpose of this review, we will focus on the socioeconomic features of the neighbourhood environment.

The socioeconomic structure of a neighbourhood encompasses historical, cultural, political, and institutional factors that work to establish the citizens’ structural position, determine the resources available, and the exposures that will affect the distribution of health outcomes in the population [15]. The neighbourhood socioeconomic structure considers the composition and the relationship between individuals who are part of the community [16], outlining the behaviours, aspirations, and social norms of neighbourhood residents [17] (p. 115). Hence, the socioeconomic environment in a community is an essential component of how people behave and define the “norm”, the patterns of social control, the engagement in specific behaviours, and the coping mechanisms used when facing adversity [18] (p. 45). Moreover, the neighbourhood socioeconomic structure is a marker for the characteristics that can affect health, including the availability of public services and environmental resources [19].

As discussed, the place we live can act as a critical social determinant of health. Therefore, this review intends to examine the role of some structural characteristics of socioeconomic deprivation and segregation in neighbourhoods. Moreover, in the current scenario where cities are in transition, the social inequities experienced by older adults at the neighbourhood level should be emphasised. Therefore, we will discuss how changes in the urban environment, which have been triggered by an unprecedented housing crisis in the past couple of years due to gentrification and urban renewal, can deepen health inequities [20].

#### 1.1.1. Neighbourhood Socioeconomic Processes and Dynamics

Ross and Mirowsky define neighbourhood socioeconomic status (also called neighbourhood socioeconomic deprivation, which is the term we will adopt in this review) as “inequality in the distribution of valued goods, resources, prosperity, and opportunity” at the macro-level [21]. These authors include three main characteristics in their definition: education, work, and economic resources. For example, the levels of education in a neighbourhood might dictate how people can express their interests by negotiating, finding and using the necessary information to improve the conditions of their neighbourhood. In addition, the number of people in a neighbourhood who are employed may settle order in the residential area. Furthermore, the levels of poverty and home ownership might dictate the economic resources of a neighbourhood. Having wealthier residents and more homeowners in the neighbourhood may create more concern for preserving and investing in the area, favouring interest in the region and, consequently, increasing its perceived quality [21]. The combination of these three dimensions is typically used to measure neighbourhood socioeconomic deprivation.

Multiple multivariate indexes are used to measure neighbourhood socioeconomic deprivation, such as the Townsend deprivation index, Carstairs deprivation index, and the Jarman or underprivileged area (UPA) score [22]. Notably, most high-income countries have their own deprivation index versions to measure deprivation in their country [23,24,25,26]. Nevertheless, generally, neighbourhood deprivation includes a composite of area aggregated measures such as the proportion of unemployed, blue-collar workers, and less educated people overall in a geographical area [22].

Neighbourhood segregation is another social feature deemed “essential” for discussion. Neighbourhood segregation is a complex process caused by the geographic separation of groups of people based on specific characteristics: organised discrimination, economic means, and individually motivated reasons [27]. In 1970, the Duncans created the dissimilarity index [28] to measure this social process. The dissimilarity index is the most frequently used measure of segregation, although nowadays, segregation is measured with multiple indexes. Overall, each of these indices try to include five dimensions of segregation: evenness (to adjust for composition), exposure (to adjust for isolation, interaction, and correlation), concentration (to adjust for the distribution of a group concerning population densities), centralisation (to adjust for closeness to the city centre), and clustering (to adjust for relative spatial proximity) [29].

According to Schelling’s tipping model, segregation can be accentuated if a previously installed group has a compositional preference in the neighbourhood. Supposing that new individuals with different characteristics move in, the initial residents may show a desire to relocate to another area. If the minority group entering the neighbourhood exceeds the tipping point for previous locals, a transitional phase for the neighbourhood may begin [27]. As the composition of the residing population changes, so does the neighbourhood environment. The socioeconomic characteristics of a population, such as the level of education, income, occupation, fertility rates, female labour force participation, housing choice, language, physiognomy, citizen involvement in community activities, religious beliefs and practices, will affect the dynamics of a neighbourhood [30]. Thus, the socioeconomic structure of a neighbourhood is not static. The mobility flow of residents of different socioeconomic classes can alter the social structure and, consequently, the dynamics of neighbourhoods.

The interplay of these characteristics reflects how neighbourhoods work. With the arrival of new dwellers, the current environment might be altered, giving rise to the so-called invasion-succession model. This model is “a theoretical construct, setting out the sequence of competitive social actions by which a human group or social activity comes to occupy and dominate a territory, formerly dominated by another group or activity.” [31]. When new individuals with other characteristics “invade” the neighbourhood, natural processes of competition for the available resources may occur. Additionally, the different views of land use or community activities might introduce some conflict between the new residents and the pre-established population. Consequently, if none of the “new” or “old” residents come to terms with each other, then one of them will eventually withdraw. If the established population prevails, then the “invasion” will be hindered, and if the newcomers dominate, then “succession” will be accomplished, and replacement of a previous well-established social group will be initiated [30].

Hence, it is perfectly natural that neighbourhoods go through some changes. In 1959, Hoover and Vernon proposed a five-stage neighbourhood life cycle model to explain the phases neighbourhoods undergo: development, transition, downgrading, thinning out, and renewal. However, it does not mean that every area will pass through all stages of growth and decline. Some may remain in a stage indefinitely, and others may rotate through several stages without necessarily going through all five of them. The transition from one stage to another will change the characteristics of residents, namely, the population density and distribution by age, race, and social status; land and dwelling use; and housing type and conditions [30].

For instance, gentrification is a transitional process where low-income neighbourhoods are modified socially—by changing the composition (e.g., level of education, income, and racial makeup) of the inhabitants—and economically, by having more extensive real estate investment [32]. Renovations in these areas, such as investments in transport and public parks, attract new residents with higher purchasing power. Any changes in land use are tailored to the new type of resident. As a result, landlords will look for a profit with these new higher-paying tenants. In return, long-term residents will notice a change in the neighbourhood’s character, and might be displaced by rent increases, evictions, and other displacement pressures as a consequence [33,34]. In fact, the previously established population may not take advantage of the new amenities. Instead, they might have to move to areas outside of their original community, placing them further away from their jobs and their family and friends, so they start to feel out of place. For the people that can afford to stay in the neighbourhood, a sense of their community may be lost. Small family businesses might also begin to disappear as their clients migrate or commercial rents are no longer affordable [32].

Another innate neighbourhood process is “filtering” due to the natural physical deterioration of housing conditions over time in the absence of proper maintenance. This process might induce higher-income individuals to seek better housing conditions and relocate to other areas [35]. As a result, lower-income people can reoccupy these areas as they become cheaper and improve their situation compared to their previous housing. However, it is important to note that these people will still be living in inadequate conditions. Creating local urban renewal programs designed to solve urban deterioration, including housing, physical infrastructure, and community services [36], might help fight the vicious cycle of poverty and inequities. The revitalisation of decaying areas with interventions such as creating or reforming public spaces, parks, and community centres can also provide better conditions for greater overall well-being. However, urban renewal programs can also lead to negative aspects related to social network disruptions, the closure of long-time establishments, and feelings of insecurity in the new environment. Another crucial aspect to consider is that an area’s renovation might alter housing prices and the overarching lifestyle that could attract newcomers to gradually generate a process of gentrification [37]. Therefore, when analysing these interventions, one should be cautious of initiating social exclusion and displacing the most vulnerable residents.

#### 1.1.2. Mechanisms Connecting Neighbourhood Structure and Dynamics and Healthy Ageing

How are social exposures connected to the mechanisms involved in the pathways of neighbourhood effects on health? According to George C. Galster, four main mechanisms explain this connection: socio-interactive, environmental, geographical, and institutional [38].

Amongst the social-interactive mechanisms [39,40], there are:processes of influence by neighbours (social contagion) where people change their behaviours, aspirations, and attitudes based on contact with their peers;processes of adhering to local social norms (collective socialisation) through neighbourhood role models or social pressures;processes of transmission of information or resources (social networks) through neighbour networks;processes of influence in the behaviour of residents through a degree of social disorder (social cohesion and control) changing also psychological reactions;processes of competition due to limited resources in the neighbourhood;processes of relative deprivation where people tend to feel inferior to their neighbourhood peers that have reached a higher socioeconomic position;and finally, the process of parental mediation where the actions of parents are influenced by the environment of the neighbourhood and, therefore, affect the way they raise their children [38].

A longitudinal analysis conducted in Germany verified that social cohesion had a mediation effect on neighbourhood environmental/built characteristics and the health of older adults, especially their physical health [41]. The authors demonstrated that social cohesion was a predictor of mental and physical health. Another longitudinal study conducted in Japan also observed that community social networks might promote participation in social activities, which in turn, might support the adaptation to a healthier diet, particularly for the elderly living alone [42].

Regarding the environmental mechanisms [43,44,45], these processes might affect individuals directly through psychological effects without necessarily changing their behaviours. When there is exposure to violence, a sense of danger may compromise the well-being of individuals. When the physical surroundings are unpleasant, residents might feel helpless and stressed. Furthermore, when there is toxic exposure to pollutants in the air, soil, or water, the health of residents will certainly be conditioned [38]. A study conducted in the United States found that higher levels of neighbourhood violence were associated with higher depressive symptoms in older adults [46]. Moreover, in a cross-sectional analysis, elderly people living in American neighbourhoods with higher exposure to fine particulate matter air pollution had worse cognitive function than their peers exposed to lower levels of air pollutants [47].

The geographical mechanisms have two main processes: spatial mismatch, when there are no appropriate opportunities for the residents’ skills; and public services, which might be restricted in a specific area, limiting the health, education, and development of residents. In other words, opportunities are limited based merely on the location of the neighbourhood [38]. For example, in China, unmet service needs felt by the elderly negatively affected their life satisfaction [48].

Lastly, the institutional mechanisms [49,50] are characterised by powerful third parties that influence residents’ lives. There is the process of stigmatisation, where certain neighbourhoods entail harmful stereotypes that can limit favourable options for the residents. Furthermore, local institutional resources and local market actors may influence access to certain services and stir up certain behaviours, respectively [38]. For instance, in the state of Pennsylvania, in the United States of America, older adults living in a neighbourhood with the highest tertile of amenity diversity had greater mobility than those who spent most of their time in their residential area [51].

Based on these mechanisms, it is not difficult to recognise that healthy surroundings are crucial for having better opportunities to thrive in life. Thus, the neighbourhood environment is central in people’s lives, particularly for the ageing population, since they are often restricted to their nearby neighbourhood environment and, therefore, more vulnerable to it.

Since the neighbourhood has such an important influence on one’s health, it is essential to acknowledge the role of social environmental factors on healthy ageing and the importance of creating age-friendly communities in the context of an ageing population. However, to the best of our knowledge, there remains missing information on how research has been conducted in this field. It is therefore essential to investigate the current evidence regarding the influence of the neighbourhood social environment on healthy ageing to guide future research. By understanding how research has been conducted, and by identifying gaps in the evidence, it will be possible to redirect our efforts and prioritise the needs in this field. Furthermore, gathering new relevant information can hasten the process of creating and optimising local and international responses to the growth in older populations and, therefore, withstand the intrinsic challenges (e.g., sustain health across advanced ages) that will prevail in the years ahead.

Our aim was to explore how neighbourhood socioeconomic processes and dynamics affect the health outcomes of older adults through a scoping review. This type of review is the most appropriate approach, since our study has a broad research question that intends to explore the extent of literature on the topic, identify key concepts in the field and sources of evidence, and clarify working definitions and the existing knowledge gaps in the literature. This scoping review is intended to emphasise two emerging societal challenges—the ageing population, and, the transitions and wide socio-spatial inequalities observed in cities—by summarising the available evidence on the topic to better understand how the social environment where older adults reside affects their health. More precisely, we aimed to understand how neighbourhood socioeconomic structure and processes influence the health of older people, specifically, neighbourhood socioeconomic deprivation, socio-spatial segregation, urban renewal, and gentrification.

## 2. Methods

### 2.1. Study Design

This scoping review followed the six-stage method developed by Arksey and O’Malley, Levac and colleagues, the Joanna Briggs Institute, and the Preferred Reporting Items for Systematic Reviews and Meta-Analyses Extension for Scoping Reviews (PRISMA-ScR) checklist (see Appendix A). The protocol of this review was not registered because scoping reviews are not currently accepted by the International Prospective Register of Systematic Reviews (PROSPERO).

Our research question is: “How do neighbourhood socioeconomic processes and dynamics affect older adults’ health?” Our research question was structured using the PCC (population, concept, context) approach to identify the components according to the 2020 Joanna Briggs Institute guidelines for Scoping Reviews [52]:

P: older adults;

C: healthy ageing;

C: neighbourhood socioeconomic deprivation, socio-spatial segregation, gentrification and urban renewal.

### 2.2. Search Strategy

Three research databases were used for this scoping review: PubMed, Web of Science™, and Scopus^®^. We combined multiple databases in the humanities and medical and social sciences to yield the best results. A literature search was performed using a search strategy developed in consultation with all authors, with no geographical or date restrictions to optimise our retrieved results. From inception to 24 April 2021, Boolean operators, truncation, and wildcards were used to refine our search, including Mesh terms in the PubMed database to identify original studies addressing the effects of neighbourhood socioeconomic processes and dynamics on older adults’ healthy ageing. All the terms used in the search expression were either applied to the title, abstract, or keywords (see Appendix A).

### 2.3. Eligibility Criteria

All studies conducted on older adults that assessed the association between neighbourhood socioeconomic processes and dynamics and older adults’ healthy ageing were included. The inclusion criteria were based on the WHO definition [8] of healthy ageing, “the process of developing and maintaining the functional ability that enables wellbeing in older age”; and on Diana Kuh and colleagues’ approach [53] that includes healthy biological ageing and well-being. The definition includes not only the “(…) maintenance (…) of optimal physical and cognitive functioning for as long as possible (…)” but also two important components: (1) the survival to old age; (2) the delay in the onset of chronic diseases and disabilities. The following successive exclusion criteria were applied: (1) article type (reviews, editorials, letters, comments, or statements); (2) non-human studies; (3) study design/content (ecological studies, cohort/data profile, or validation of a questionnaire); (4) studies in languages other than English, French, German, Italian, Portuguese, or Spanish; (5) studies without access to full-text; (6) studies without a comparison group; (7) studies where the population is composed of patients or other non-healthy populations exclusively; (8) studies that do not have information on individuals aged 60 years (cut-off for old age by the United Nations [54]) and older; (9) studies not reporting the age of the studied population; (10) studies without an exposure measure of the neighbourhood socioeconomic processes and dynamics such as neighbourhood socioeconomic context, spatial segregation, gentrification, or urban renewal; (11) studies where the exposure was verified during childhood only; (12) studies without a measure of healthy ageing; and (13) studies that do not provide information on the associations of neighbourhood socioeconomic processes and dynamics with healthy ageing.

### 2.4. Selection of Studies

After exporting all the references identified in the three databases (*n* = 7100) to an EndNote Library, duplicates (*n* = 2611) were removed automatically with the tools provided by the software. Additionally, references were also checked manually to remove any duplicates not previously identified (*n* = 259). Thus, of the 7100 references exported, 2870 were duplicates (40.4%), leaving a total of 4230 final references for analysis. References were independently screened based on the title and abstract according to the authors’ predefined inclusion and exclusion criteria. The authors discussed any disagreement to reach a consensus in all cases. The papers selected by the screening phase (*n* = 684) were assessed for potential eligibility for full-text analysis, and 115 studies were included in the analysis. Backward and forward citation tracking was also conducted by screening the reference list of the included studies and looking for papers that cited them. Twelve studies were identified as eligible by backward citation tracking, but only seven studies fulfilled the predefined criteria and were included. In the end, 122 studies were included in the review (see Figure 1).

### 2.5. Data Extraction

Considering the studies that met the inclusion criteria (*n* = 122), reviewers independently extracted data into a spreadsheet developed by the research team to collect all relevant information. Data on the authors of the study, year of publication, the country, study design, participants’ age, sample size, exposure and its measurement, healthy ageing outcome, and its measurement, as well as the estimated effect obtained between the exposure of interest and the outcome, were extracted independently (see Appendix A). Healthy ageing outcomes were grouped into broader categories defined by the authors based on the conditions and areas of health (see Appendix A).

## 3. Results

### 3.1. General Characteristics of the Included Studies

One hundred and twenty-two publications were included. The studies were published between 2001 and 2021, totalling 20 years of research (see Figure 2). Four of these studies combined data from different countries, namely, Belgium and China, and Belgium and England; and two studies resulted from a partnership between Slovakia and Netherlands. North America produced more scientific evidence on these topics than any other continent, with a total of sixty-four studies from the United States of America (*n* = 61) and Canada (*n* = 3). Thirty-six studies were conducted in Europe (United Kingdom (*n* = 21), the Netherlands (*n* = 4), Spain (*n* = 2), Sweden (*n* = 3), France (*n* = 2), Germany (*n* = 1), Denmark (*n* = 1), Portugal (*n* = 1), Switzerland (*n* = 1)); eight studies in Asia (China (*n* = 4), Singapore (*n* = 2), South Korea (*n* = 1), Vietnam (*n* = 1)); four studies were conducted in South America (Brazil (*n* = 4)); four studies in Oceania (New Zealand (*n* = 3), Australia (*n* = 1)); and a single study in Africa (Ghana (*n* = 1)) (see Figure 3). The sample size of the studies ranged from 63 to 4 526 759 individuals, and the age range of the studies was between 60 and 111. The most common study designs used in this specific field were cross-sectional studies (*n* = 71), followed by longitudinal studies (*n* = 41), qualitative studies (*n* = 6), mixed-methods studies (*n* = 2), quasi-experimental studies (*n* = 1), and case-control studies (*n* = 1).

### 3.2. Definition and Measurement of Neighbourhood Socioeconomic Processes and Dynamics

#### 3.2.1. Geographical Unit of Analysis

Neighbourhoods were defined based on different geographical scales, and their use varied among countries. Most studies used administrative, statistical, or census geographical units of analysis (*n* = 118), and a few used ego-centric definitions based on fixed distance buffers (*n* = 4). Although most of the studies did not report such information (*n* = 87), among those that did, the number of inhabitants per unit of analysis was between 559 and 240,000, the size was between 0.4 and 228 km^2^, and the number of participants per unit area was between 1 and 170.

#### 3.2.2. Exposures

From the included studies, four groups were established to measure the exposure of neighbourhood socioeconomic processes and dynamics, namely neighbourhood socioeconomic deprivation (*n* = 114), gentrification (*n* = 5), segregation (*n* = 2), and urban renewal (*n* = 1).

##### Gentrification

Except for one quasi-experimental study, the remaining papers evaluating gentrification were qualitative. Two studies were conducted in the United States of America, one in the United Kingdom, one in Spain and another in Canada. All five studies were published in the past ten years, and four were published in the last four years, suggesting that more attention has been given to the topic or that gentrification is now more widespread. The definition of gentrification was relatively consistent between studies. Gentrification was defined as a transitional stage where low-income neighbourhoods become wealthier. The conceptualisation of the term was based on the transformation of neighbourhood infrastructure, neighbourhood composition changes, and a rise in real estate prices, services, and goods. The operationalisation of gentrification was assessed by comparing “before” and “after” the transformation of neighbourhoods; in other words, how the area evolved over the past decades. This information was based on census information [55,56]; however, in some studies, it was not clear where the data were retrieved [57,58]. The measures used to describe this phenomenon included an increase in white-collar workers, younger people, higher educated and wealthier people, ethnic/racial changes, rise in property prices, change in land use with an increase in new apartments or houses, and contemporary amenities for a new lifestyle, for instance, wine bars, coffee shops, and vegetarian cafes [55,56,57,58]. One of the studies focused on the gentrification process of a medium-sized city’s historical centre due to tourism. The study area had become increasingly tourist-orientated, altering the urban environment to suit the needs of tourists. The operationalisation of gentrification in this study was based on official reports, the creation of tourist-related activities, tourism sector awards, and the policies created to maintain and promote tourism [59].

##### Segregation

The only country that researched this exposure was the United States of America. The two studies conducted were cross-sectional studies. The type of spatial segregation evaluated was either by race (*n* = 1) or age (*n* = 1). Racial segregation was measured using the dissimilarity index [60]. Morrill’s spatially segregation index—an adaptation of the Duncan and Duncan spatial index that considers the contiguity among spatial units—was used to measure spatial age segregation [61].

##### Urban Renewal

Only one qualitative study, conducted in the Netherlands, investigated urban renewal. As reported by the authors, the Netherlands has state-led urban renewal strategies for disadvantaged urban neighbourhoods. Thus, a Northern Netherlands neighbourhood with 11,575 residents (8% aged ≤ 65) undergoing an urban renewal process was chosen. Initially, in 1998, the municipality and housing corporations aimed to attract middle-class families to improve the social and physical character of the area. Later on, in 2007, these entities started focusing on social neighbourhood renewal to increase resident participation and social cohesion [62].

##### Neighbourhood Socioeconomic Deprivation

Regarding the type of study, most were cross-sectional (*n* = 69), followed by longitudinal (*n* = 41), mixed-methods (*n* = 2), one qualitative study (*n* = 1), and one case-control study (*n* = 1). The way neighbourhood socioeconomic status/deprivation was measured varied differently among the studies. For example, some studies used only one measure to evaluate the neighbourhood’s socioeconomic status, such as registered unemployment [63,64] and income [65,66,67,68], proportion of social welfare recipients [69], education [70,71], poverty [72,73,74], and the percentage of adults aged 25 or older [75]. Most studies used multiple indicators to calculate deprivation indices that varied with the type of measures used. These indices were constructed from a combination of sets based on education, income/wealth, (un)employment, poverty, household tenure/composition/density, housing, occupation, living environment, barriers to housing and services, health deprivation and disability, households receiving public assistance, crime or fear of crime, demographic information, racial or ethnic heterogeneity/group, skills and training, residential stability, and sense of belonging and trust in people (see Figure 4 and Appendix A). One study did not specify the indicators used.

#### 3.2.3. Measurement of Healthy Ageing

There were quite a few different outcomes related to healthy ageing, namely, neurological health (*n* = 44), social/physical functioning (*n* = 14), well-being/quality of life (*n* = 14), health behaviours (*n* = 13), self-rated health (*n* = 10), cardiovascular health (*n* = 9), endocrine, nutritional, and metabolic health (*n* = 8), mortality (*n* = 7), ageing in place (*n* = 2), life expectancy/years of potential life lost (*n* = 2), access to health (*n* = 2), oral health (*n* = 1), multimorbidity (*n* = 1), and muscular skeleton system (*n* = 1).

The most researched item—neurological health—included outcomes such as Alzheimer’s disease, anxiety/depression and its symptoms, psychosocial functioning, cognitive decline and impairment, dementia, mental health, and cognitive function.

The social/physical functioning domain included outcomes such as functional status/limitation/impairment, mobility impairment, social and physical functioning, falls, physical health/function, mobility disability, disability, and frailty.

The well-being/quality of life category explored themes such as anger, health conditions, loneliness, overall well-being, psychological distress, quality of life, shopping difficulty, and social exclusion.

The themes addressed in the health behaviours category were diet, physical activity and exercise, health-(risk) behaviours, and drinking problems.

Cardiovascular health included outcomes such as cardiovascular disease, coronary heart disease, hypertension, stroke, and ischaemic stroke.

The outcomes explored in the endocrine, nutritional, and metabolic health group were body mass index, diabetes, metabolic conditions, obesity, chronic inflammation, and progressive chronic kidney disease.

Regarding mortality, it was verified by all-cause mortality, or by specific causes such as cardiovascular disease, ischaemic heart disease, or stroke.

There were also studies measuring ageing in place, access to healthcare, life expectancy/years of potential life lost, and self-rated health.

Finally, there was one study for each of the following categories: musculoskeletal system, oral health, and multimorbidity (Figure 5).

#### 3.2.4. Associations

##### Gentrification

The only quantitative study showed that economically vulnerable older adults living in gentrifying neighbourhoods had higher scores for self-rated health than those in low-income neighbourhoods. However, economically vulnerable and higher-income elders had poorer mental health (more symptoms of anxiety and depression) than individuals living in non-gentrifying neighbourhoods [57]. The remaining qualitative studies were valuable sources of information on how older people felt and dealt with gentrification. Overall, older adults felt socially excluded from their neighbourhood [55,56] and recognised that its transformation made them lose their social places, neighbourhood identity, and community cohesion [58]. Other common reported opinions were loneliness, disconnectedness from the neighbourhood [56,59], and loss of political power [56]. Moreover, some older adults noticed a price increase in commerce, which meant they had to carry out their activities further away from their home area, and thus, had to be more dependent on family members [59]. Inflation in housing also induced displacement worries [58]. However, older people highlighted some positive notes about gentrification: the construction of an informal network of neighbours that support each other [55], their capacity to adapt to change based on successful past experiences, and the benefits of neighbourhood change, namely, the added infrastructure and intergenerational programs [58]. The neighbourhood exposures chosen to explore the underlying mechanisms related to neighbourhoods and healthy ageing were focused on the environment’s physical features, namely, the transformation of neighbourhood infrastructure and the change in land use.

##### Segregation

Self-rated health was the only category within healthy ageing that was analysed by the studies. There was no substantial evidence for an association between segregation and self-rated health [60,61]. However, in the spatial age segregation study, there was a marginal increase in the number of days of “mental unhealthiness” (the average number of days where mental health was not good during the past 30 days) [61]. These two studies found that the mechanisms underlying the associations were related to the social features of the neighbourhood, namely, racial segregation, racial composition, percentage of households in poverty, and voluntary residential selection processes.

##### Urban Renewal

Only one qualitative study analysed the effect of urban renewal on ageing in place. Similar to gentrification, older people recognised that renovation made them lose their neighbourhood identity and places for their social encounters [62].

##### Neighbourhood Socioeconomic Deprivation

Regarding access to health, the two studies that analysed this outcome had a different approach: the longitudinal study assessed who had more visits to the emergency department, and the cross-sectional study assessed who had more health access problems. Nevertheless, in both scenarios, older adults living in poorer neighbourhoods had higher rates of ambulatory visits and more difficulty accessing health [76,77]. Furthermore, neighbourhood ethnic composition seemed to be important for these outcomes, since higher rates of ambulatory visits were observed for elderly Hispanic residents, and heterogeneous neighbourhoods had more problems accessing health [77].

No strong evidence was found that neighbourhood deprivation was associated with physical functioning or health-related quality of life in the cross-sectional study [69].

Regarding cardiovascular health, living in low-educated [70] or more disadvantaged neighbourhoods was associated with having hypertension [78]. However, one study did not find an association between neighbourhood socioeconomic deprivation and hypertension [79]. Another three studies did not find an association for ischemic stroke [80], subclinical cardiovascular disease [81], or self-reported physician-diagnosed cardiovascular disease [82]. Furthermore, neighbourhood deprivation was adversely related to coronary heart disease [83,84] and stroke [85].

With respect to endocrine, nutritional, and metabolic health, living in more disadvantaged neighbourhoods was associated with higher odds of chronic inflammation [86], obesity [65,87,88], and progressive chronic kidney disease [89]. Moreover, no association was observed between neighbourhood socioeconomic deprivation and diabetes in a cross-sectional study [90]. However, one longitudinal study observed that living in a less deprived neighbourhood was associated with an increased risk of diabetes [91]. A longitudinal study verified that older adults living in a less deprived neighbourhood was associated with having a healthier body mass index at baseline and protected against age-related weight loss over time [92].

Concerning health behaviours, high neighbourhood socioeconomic deprivation was positively associated with having a higher intake of fruit and vegetables [93]. However, when assessing two different countries—Netherlands and Slovakia—low consumption of fruits and vegetables was not associated with neighbourhood unemployment (used as the measure of area deprivation), binge drinking, or a lack of physical activity. In this same study, older Dutch people living in the least favourable neighbourhoods were more likely to smoke daily and be overweight [63]. In one study conducted in the United Kingdom, elderly residents living in more deprived areas ate fewer fruits and vegetables, as well as less chicken and fish. In the same study, it was also observed that people living in those deprived areas smoked more and lacked regular physical exercise compared to those living in more affluent areas [94]. In addition, when considering neighbourhood ethnic composition, older Mexican American men living in a neighbourhood with a higher Mexican American density had fewer drinking problems [95]. Area socioeconomic deprivation was unrelated to within-neighbourhood walking and within-neighbourhood walking for recreational purposes [96]; however, other studies found that neighbourhoods with the lowest income had the highest physical inactivity [97,98] or affected the elderly’s leisure-time physical activity [99]. In a mixed-method study, older adults living in less deprived neighbourhoods perceived that their neighbourhood facilitated active leisure. In contrast, older adults living in more deprived neighbourhoods had the opposite perception [100]. One study found that older adults living in medium-income areas were more likely to walk for leisure [101]. Another study verified that home ownership and occupancy were positively associated with neighbourhood walking [102]. However, some studies did not find an association between neighbourhood deprivation and physical activity [103] or leisure-time physical activity [104].

Regarding life expectancy/years of potential life lost, higher socioeconomic neighbourhoods had a higher life expectancy [105], and the likelihood of becoming a centenarian was lower in more deprived neighbourhoods [106].

With respect to multimorbidity, no difference in the risk of multimorbidity was observed across quintiles of area deprivation index, after the age of 70, unlike what was seen for younger ages [107].

Among studies that evaluated mortality, all results indicated that living in more deprived neighbourhoods was associated with a worse mortality prognosis [75,108,109]. For instance, living in poorer neighbourhoods was associated with cardiovascular mortality [110,111,112]. In addition, one study verified that higher mortality was associated with individuals who lived in areas with many unemployed inhabitants [113].

With regards to the musculoskeletal system, older adults living in more deprived neighbourhoods had lower odds of suffering a hip fracture [114].

In relation to neurological health, people living in the most disadvantaged neighbourhoods had higher odds of Alzheimer’s disease [115]. Another study found no substantial evidence that older adults living in deprived neighbourhoods were associated with anxiety or depression [116]. Psychosocial functioning was inversely associated with neighbourhood socioeconomic deprivation [117]. Concerning cognitive function, living in socioeconomically disadvantaged neighbourhoods was associated with worse cognitive function [71,118,119,120,121]. However, some studies did not find a significant association [68,122,123]. Living in less affluent areas was associated with developing cognitive impairment [124,125,126,127] as well as cognitive decline [67,128,129,130], and cognitive function [131], dementia [132,133,134], and depression [74,118,135,136,137]. Moreover, living in neighbourhoods with less residential stability [138] was positively associated with depression and, although the association was not significant, residing in an area that underwent an increase in the percentage of poor increased the odds of developing depressive symptoms [72]. However, two studies did not find a significant association between neighbourhood socioeconomic deprivation and cognitive decline [139] or mental health [64,140]. Depressive symptoms also had the same association, where poorer neighbourhoods had worse results [141,142,143] or no association [144,145]. However, in general, living in deprived neighbourhoods was associated with worse mental health [66,146,147,148,149].

Regarding oral health, older adults living in the most deprived areas only used dental services when symptomatic [150].

Concerning self-rated health, older adults are more likely to report poor health if living in more deprived neighbourhoods [82,151,152,153,154,155,156], or when they lived in an area with a lower education level [151].

Among social and physical functioning, older adults living in less affluent neighbourhoods had higher odds of reporting a functional impairment [127] or limitation [157,158,159,160]. However, in another study, the same association disappeared after adjusting for individuals’ characteristics [161]. Living in more deprived areas was associated with being more vulnerable to being frail [162,163,164] and having higher rates of falls [165,166]. In addition, older adults living in poorer neighbourhoods had worse physical health perceptions [148]. There was no association between living in disadvantaged neighbourhoods and functional status in another study [82].

Regarding well-being/quality of life, living in a disadvantaged neighbourhood was positively associated with anger among both lower-income older adults who felt financially advantaged compared to their neighbours and higher-income elders who felt economically disadvantaged compared to their neighbours [167]. Disadvantaged neighbourhoods also contributed to shopping difficulty [168], worse health [169], and loneliness [170]. However, there was no strong evidence that neighbourhood characteristics were associated with loneliness in another study [171]. Reduced odds of psychological distress was associated with greater employment deprivation [172]. Regarding the quality of life, there was no statistically significant difference between slum and non-slum respondents for physical and psychological quality of life. However, slum respondents reported significantly higher social quality of life, and non-slum respondents reported significantly higher environmental quality of life [173]. In another study, higher neighbourhood deprivation was significantly associated with lower self-perceived quality of life in the physical and environmental domains [174]. Again, living in a poorer neighbourhood resulted in a lower quality of life [175]. Older adults felt excluded by the changing composition of their locality, felt insecure about criminal activity in their locality, avoided certain public spaces (especially women), and coped with some of their difficulties by making risk management strategies [176].

With regards to deprivation, several neighbourhood exposures were included to understand the mechanisms behind it, namely, the environment’s shared social and physical features; the availability of (un)favourable home, work, and leisure environments; services to support daily living; sociocultural features; and neighbourhood reputation.

Figure 6 provides a synthesis of the evidence from all of the studies included in this scoping review in the form of a harvest plot.

## 4. Discussion

This scoping review on the influence of neighbourhood socioeconomic processes and dynamics on healthy ageing indicates that the research has been focused on neighbourhood socioeconomic deprivation, with a limited number of studies dedicated to analysing the effects of socio-spatial segregation or the processes of neighbourhood change, such as gentrification and urban renewal.

We found that, in general, older people living in deprived neighbourhoods had worse healthy ageing outcomes than their counterparts living in more advantaged neighbourhoods. However, a few studies (*n* = 33) did not find a significant association, and four studies found the opposite association. The results differed according to the type of study, the geographical area, and the outcome measures analysed. There are various possible explanations as to why there were mixed findings. One reason might be related to the use of different measures of neighbourhood deprivation: some studies relied on univariable measures, whereas others relied on multivariable indexes, which, in principle, are more adequate for grasping the multidimensional construct of deprivation. In addition, there is evidence that the effects of neighbourhood deprivation may depend on macro- and country-level features; for instance, it has been demonstrated that individuals in more egalitarian countries may be less spatially segregated and thereby less affected by the harm of living in certain neighbourhood environments than their counterparts living in liberal unequal regimes [19,177]. Another possible explanation is that sample distribution in the geographical units can compromise the power to detect an association or even the disposition of neighbourhood deprivation in the selected area. Moreover, the set of confounding variables used to estimate adjusted associations varied widely between studies (for instance, 89% adjusted for no individual-level socioeconomic factors and 11% adjusted for one only). Disregarding important covariates may lead to residual confounding, but over-adjustment may lead to biased results as well. The chosen geographical unit of analysis might be another explanation for these variations. Although most studies used administrative/statistical/census areas to serve as proxies for the neighbourhood—for the ease of compiling the aggregated information—the results may be misleading if they do not reflect the real geographical distribution of the social environment or the residents’ subjective neighbourhood boundaries [178]. In addition, the area size chosen to analyse the neighbourhood social context is crucial. The selected scale might alter the results due to the modifiable areal unit problem. Restricting or amplifying the geographical area may belie the study results by excluding certain hazards and resources or masking relevant variation, respectively [179]. Moreover, the administrative definitions of census blocks or tracts, counties, districts, postal codes or other geographical units used in the studies differ in size depending on the country, and even within a country, contributing to the added variation in the measurement of the social environment. Since neighbourhood definitions are operationalised differently, comparisons of findings between studies can be more demanding. Another vital characteristic to be considered is the correct selection of the spatial unit to analyse the social environment. If the selected area is too heterogeneous, or even too homogeneous, it will compromise the results for the contextual effects on the outcome being analysed [178].

Another crucial aspect to consider is that neighbourhood socioeconomic deprivation was measured differently across the included studies. For example, some studies assessed area-level social variables based on one measure, and others were based on multiple indicators. The completeness of the information to determine this exposure can be related to the study’s objective, if it was the primary exposure or, for instance, explored as a control variable. The heterogeneity observed in measuring these exposures may contribute to the discrepancies observed in the associations; therefore, a more robust conceptualisation is needed.

Our results also found that socio-spatial segregation was explored in only two studies conducted in the United States of America. The emphasis on segregation in this country might be related to historical reasons and its relevance to health disparities (in race) [180]. However, the type of segregation was measured differently between these studies. It will be essential to further explore the segregation dimension in other countries, since different countries may have different degrees of socio-spatial segregation, not only by race, but also by socioeconomic positioning and education levels, for example [181]. Moreover, it is an essential social feature to consider since there is no random distribution of people in a neighbourhood, and because high levels of segregation may accentuate socio-spatial injustices in the distribution of health conducting community resources. Lastly, it would be relevant to explore the effect of segregation on other healthy ageing outcomes as both studies analysed self-rated health.

Regarding the definition of urban renewal and gentrification, it was quite consensual between studies. What was also consistent were the opinions of older adults regarding these processes. In both neighbourhood processes, older adults pointed out more negative aspects than positive ones. Thus, when these neighbourhood changes occur, it would be essential to consider the neighbourhood composition to ensure that the characteristics preferred by the population are respected and, in this way, contribute to ageing in place. It would also be relevant to analyse the effects of social changes across the life course for future research, as most research is cross-sectional and qualitative. The quasi-experimental study indicated that, although living in gentrifying neighbourhoods compromises mental health outcomes, it was not the case for the self-rated health of economically vulnerable elderly residents. The benefit of these elders’ self-rated health might be related to better amenities and services provided by gentrifying neighbourhoods. This situation might be explained by environmental mechanisms involved in the pathways of neighbourhood effects on healthy ageing, whereas the negative feelings towards gentrification might be related to social-interactive mechanisms.

Furthermore, some associations between the social neighbourhood environment and healthy ageing were moderated by more individual characteristics, such as age, sex, ethnicity/race, level of educational attainment, occupation, unemployment, and socioeconomic status, or by neighbourhood characteristics, such as length of residence in the neighbourhood, residential stability, and neighbourhood heterogeneity. These variables are important factors to consider since they can influence susceptibility to some detrimental effects of the social neighbourhood environment.

Another gap verified by this review is that research regarding neighbourhood change processes such as urban renewal and gentrification has been conducted mainly using a qualitative approach. Although evidence has been collected on how older people are experiencing the changes in their neighbourhoods, there is still little evidence in the form of quantitative data. Interestingly, in gentrification, a quasi-experimental study was performed. The neighbourhood modification phenomenon is actually a natural process that can be more easily explored as a natural experience.

The effect on healthy ageing due to the change in the neighbourhood social environment was not extensively explored by the longitudinal studies. Most studies analysed the effect of healthy ageing in a relatively stable neighbourhood social environment during the analysis period. Only one study evaluated how a change in the percentage of poor in the neighbourhood affected depressive symptoms [72]. In addition, both studies that assessed the effect of segregation on healthy ageing were cross-sectional. It would be crucial to have more information on how this exposure changes over time.

Moreover, most studies have been conducted in high-income countries, and less information is known about healthy ageing and the neighbourhood social environment in low- and middle-income countries.

Finally, most studies did not explore the potential mechanisms underlying the observed associations. Obtaining a clearer understanding of the pathways by which neighbourhood social structure and dynamics exert their effects is essential for public policy formulation [181]. Amongst the studies that addressed the underlying pathways, we observed that the most relevant mechanisms were: geographic mechanisms, namely, the limitation on meeting the health services needs of the population [137,150]; institutional mechanisms, such as local institutional resources and local market actors that influence some behaviour patterns of individuals [94,101,155]; and also social-interactive mechanisms such as social contagion [84,94] or social cohesion and control [72,118,135,140,160,168], where some behaviours are influenced by acceptance and contact with the neighbours. Environmental mechanisms such as the physical surroundings seem to influence how people conduct their choices and actions [66,84,88,94,96,101,106,149,160,168]. Thus, neighbourhood-based solutions should target these features especially.

### 4.1. Strengths

The strengths of this scoping review relied on our comprehensive literature search strategy that included three electronic databases, with no date restrictions, and a set of synonyms expressions to expand our research results that would not exclude any potential eligible articles. A clearly described search strategy was provided to increase the transparency of our results. Moreover, the inclusion/exclusion criteria were defined a priori by all authors and were followed strictly and independently reviewed in pairs, which contributed to the accuracy and reliability of the data included. Additionally, backward and forward citation tracking was performed considering the inclusion criteria to ensure no potential article was excluded from the analyses.

### 4.2. Limitations

This literature review, however, also presents some limitations that should be mentioned. After experimenting with the research terms in all fields of the three databases, the authors found that many of the retrieved results were not related to the research question itself, capturing unwanted results. Thus, we opted to restrict the research terms to the title, abstract, and keywords to improve the “positive predictive value” of the studies retrieved. However, we consider that this decision did not compromise our review excessively. We also restricted our search to the dominant languages, which could exclude relevant studies and compromise this study’s conclusions. Finally, the authors did not search for grey literature or unpublished papers due to the difficulty in systematically searching for them; however, due to the theme of this scoping review, the authors believe that additional eligible articles would not be found through this technique.

## 5. Conclusions

In conclusion, despite the large heterogeneity in measurements and designs, the literature suggests that a poor neighbourhood social environment negatively influences healthy ageing outcomes. Our study also revealed that there is still room for exploring the effects of socio-spatial segregation, gentrification, and urban renewal on healthy ageing since these exposures were not extensively analysed in the literature. Most studies on gentrification and urban renewal had a qualitative design, whereas those on segregation all adopted a cross-sectional analysis. Other observational studies, particularly longitudinal, should also be conducted to draw better inferences on the effects of the neighbourhood social environment on healthy ageing. In addition, most of the findings are correlational and disregard other potential causal factors. This calls for well-formulated theories and frameworks backed up by biological, environmental, and social sciences. In addition, when completing these assessments, it is essential to consider the way neighbourhoods/exposures are being defined. While there was a rather clear working definition for gentrification, there was a great deal of diversity in defining neighbourhood deprivation. In the current scenario of an ageing population and the transitions observed in cities with wider socio-spatial inequalities, it is essential to advance promptly and recognise that particular neighbourhood social features impact certain groups differently. Namely, neighbourhood renovations, which are happening all over the world to improve the quality of life, must be closely monitored to ascertain their potential negative and positive impacts with special consideration for the interests and resources of older adults so that they can continue enjoying ageing in their chosen place of residence.

## Figures and Tables

**Figure 1 ijerph-19-06745-f001:**
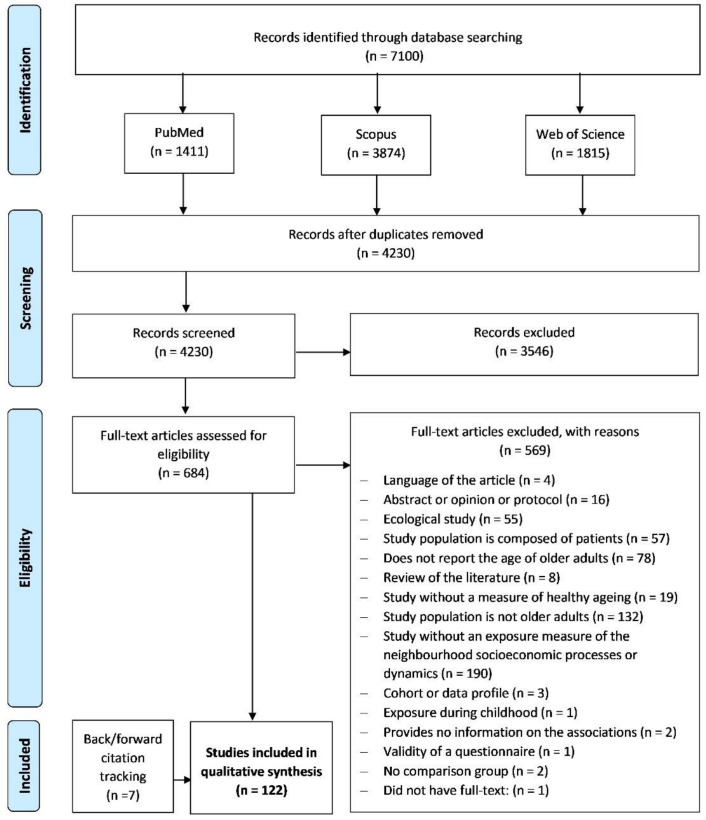
Systematic review flow-chart.

**Figure 2 ijerph-19-06745-f002:**
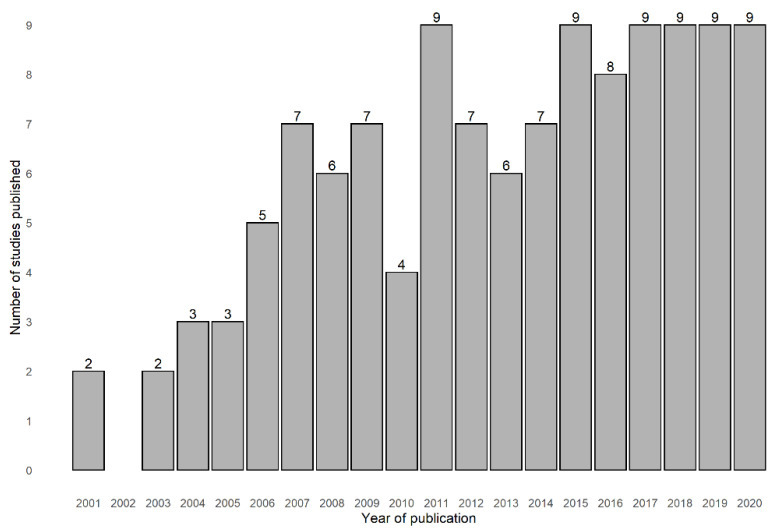
Distribution of the number of studies that assessed the association between healthy ageing and neighbourhood socioeconomic processes and dynamics by year of publication (2001–2020). The year 2021 was not included since the period was not a complete year.

**Figure 3 ijerph-19-06745-f003:**
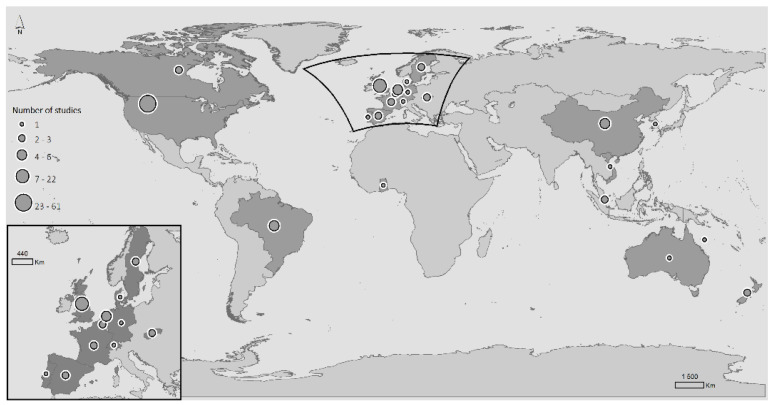
Distribution by country of the number of studies that assessed the association between healthy ageing and neighbourhood socioeconomic processes and dynamics.

**Figure 4 ijerph-19-06745-f004:**
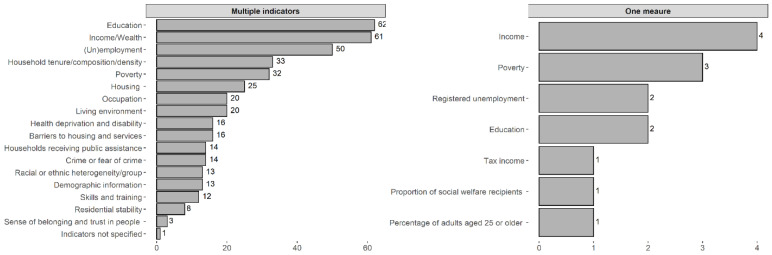
Distribution of the indicators used to measure neighbourhood socioeconomic deprivation.

**Figure 5 ijerph-19-06745-f005:**
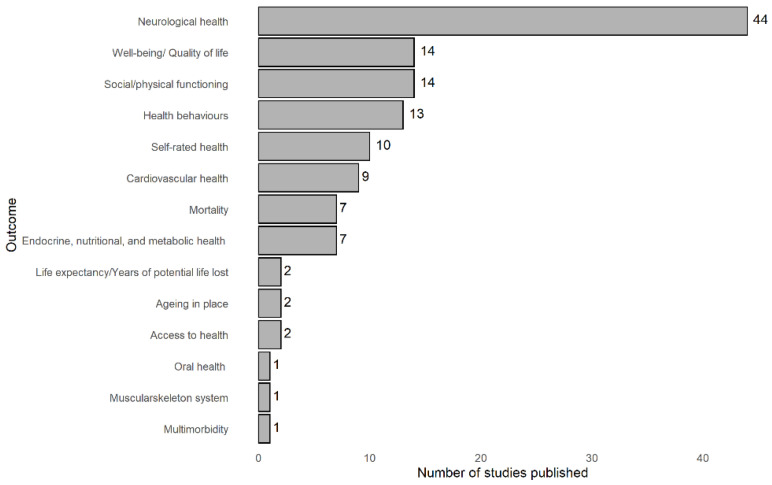
Distribution of the healthy ageing outcomes used in the studies analysed.

**Figure 6 ijerph-19-06745-f006:**
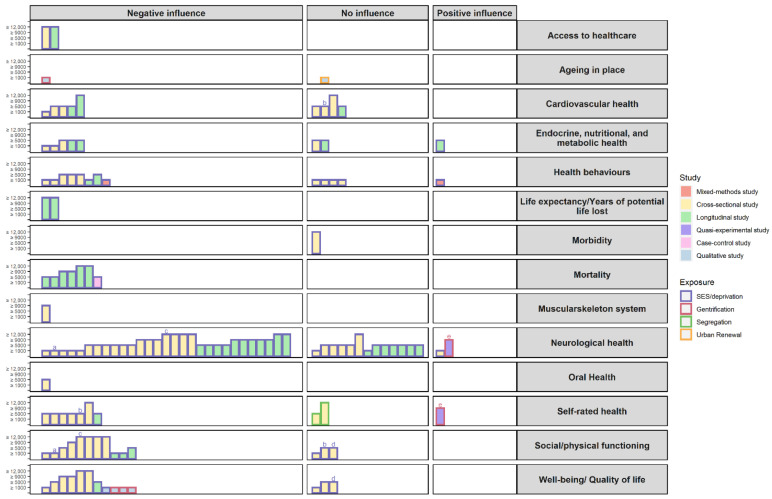
Harvest plot summarising the features of the studies included. Each bar represents a single study; however, each study can appear more than once if the authors analysed more than one outcome of the healthy ageing categories; in those cases, the bars are identified with the same letter above it (a = Kwag et al., 2011, b = Wight et al., 2008, c = Basta et al., 2007, d = Vogt et al., 2015, e = Smith, Lehning, and Kim 2018). The height of the bars is proportional to the number of participants in the studies. The colour fill of the bars represents the type of study used and the colour of the contour of the bar represents the neighbourhood exposure.

## Data Availability

Not applicable.

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
