# Peer review of "Neighbourhood Socioeconomic Processes and Dynamics and Healthy Ageing: A Scoping Review"

_ijerph, 2022, doi:10.3390/ijerph19116745_

Round 1
Reviewer 1 Report
The study is very interesting and well-conducted.
While there is quite a heterogeneity in the data sources, thus a limitation recognized as such by the authors who de facto restricted their search terms accordingly, the significant data is well analyzed and synthesized in conclusions.
The only minus is the conclusions themselves. The authors could elaborate a bit more on the findings, as much analysis leads to a sort of pre-conceived conclusion that was transparent since the hypothesis stage.
Lastly, a minor English error to be corrected on line 796: "dominant" instead of "dominated".
Good luck!
Author Response
"Please see the attachment."

Reviewer 2 Report
This paper significantly contributes to the literature by providing a rigorous and well justified systematic review of neighbourhood factors that may impact healthy aging.
The paper is very well written, providing a clear and strong narrative review to establish the rationale for their study. The method is very well conducted and described, following PRISMA-SR steps and principles. The discussion is well considered and articulated and reasonable comparisons and conclusions are drawn.
The authors acknowledge the potential limitations with their keyword search, and I agree with them that their approach was reasonable and a good representation of published peer-reviewed papers were included. I assume that this scoping review is a component of a larger study and I am sure the authors will review grey literature for that larger study.
I would like to make three suggestions to improve the clarity and rigor of the paper.
- Provide a rationale for the exclusion of cohort studies
- Line 689 - this is a general discussion of factors. Consider strengthening this discussion point by referring directly to the methods of the 23 studies that might have led to the mixed findings.
- Conclusion - please also reiterate your observation that the findings are mostly correlational and studies do not account for other potentially causal factors.
Congratulations on the paper.
Author Response
"Please see the attachment."

Reviewer 3 Report
The paper entitled "Neighbourhood Socioeconomic Processes and Dynamics and Healthy Ageing: A Scoping Review" studies an important topic. Some aspects should be improved:
1. In the abstract, it is important to briefly describe the implications of the main results found.
2. In the introduction, the authors must expand the contribution of the study. Why is it important to do this study? What gap is covered by this research in the literature on this topic? what do these results imply?
3. I believe that the methods applied should extend the justification of the authors.
4. In the conclusions, it is necessary to expand them. What do these results imply?
Author Response
"Please see the attachment."

Round 2
Reviewer 3 Report
I consider that the authors have improved the new version of the paper and it could be published.